

# Honey bees communicate distance via non-linear waggle duration functions

Patrick L. Kohl and Benjamin Rutschmann

Department of Animal Ecology and Tropical Biology, Biocenter, University of Würzburg, Würzburg, Germany

Corresponding author
Patrick L. Kohl,
patrick.kohl@uni-wuerzburg.de

## ABSTRACT

Honey bees (genus *Apis*) can communicate the approximate location of a resource to their nestmates via the waggle dance. The distance to a goal is encoded by the duration of the waggle phase of the dance, but the precise shape of this distance-duration relationship is ambiguous: earlier studies (before the 1990s) proposed that it is non-linear, with the increase in waggle duration flattening with distance, while more recent studies suggested that it follows a simple linear function (i.e. a straight line). Strikingly, authors of earlier studies trained bees to much longer distances than authors of more recent studies, but unfortunately they usually measured the duration of dance circuits (waggle phase plus return phase of the dance), which is only a correlate of the bees' distance signal. We trained honey bees (*A. mellifera carnica*) to visit sugar feeders over a relatively long array of distances between 0.1 and 1.7 km from the hive and measured the duration of both the waggle phase and the return phase of their dances from video recordings. The distance-related increase in waggle duration was better described by a non-linear model with a decreasing slope than by a simple linear model. The relationship was equally well captured by a model with two linear segments separated at a "break-point" at 1 km distance. In turn, the relationship between return phase duration and distance was sufficiently well described by a simple linear model. The data suggest that honey bees process flight distance differently before and beyond a certain threshold distance. While the physiological and evolutionary causes of this behavior remain to be explored, our results can be applied to improve the estimation of honey bee foraging distances based on the decoding of waggle dances.

## INTRODUCTION

Honey bees can use the waggle dance to communicate the approximate locations of resources to their nestmates (*von Frisch, 1965*; *Dyer, 2002*). Upon returning from a successful trip a worker honey bee may perform sequences of dance circuits on the comb alternating to the left and to the right. Each circuit is composed of the waggle phase, in which the bee walks straight and shakes its abdomen, and the return phase, in which it turns back and walks to the starting point. It is the waggle phase of the dance which conveys the information about the direction and the distance of the resource in the field (*Michelsen et al., 1992*; *Dyer, 2002*). A foraging honey bee uses the sun's position or the

polarization patterns of sunlight on the blue sky to obtain compass information (*von Frisch, 1965*). In the dance, the direction to the resource in relation to the nest is encoded by the bee's body orientation during the waggle runs. When dancing on a horizontal surface in the open, a bee refers to celestial cues and directly "points" its dances towards the goal. When dancing on the vertical comb inside the dark hive, a dancer refers to gravity, and its body orientation with reference to the upwards direction encodes the direction to the resource with reference the sun's current position over the horizon. Provided with information of the context of a dance, a human observer can unambiguously infer the compass direction indicated by a dancer of any honey bee species (*Lindauer, 1956*; see *Dyer, 1991*, *2002* or *Preece & Beekman, 2014* for reviews of direction encoding in the waggle dance). The interpretation of the bees' distance signal is less straightforward. Foragers visually gauge the length of the flight path to the goal based on the amount of image motion perceived during flight (optic flow) and encode it in the duration of the waggle phase, with longer waggle phases indicating further distances (*Esch & Burns, 1996*; *Srinivasan, 2000*; *Esch et al., 2001*; *De Marco & Menzel, 2005*). However, there is no universal distance-code and different (sub)species of honey bee can vary substantially in the slopes of their distance-duration functions (*Lindauer, 1956*; *Bosch, 1957*; *Kohl et al., 2020*). Apart from that, it is not even clear what is the precise shape of the distance-duration relationship. The current view is that waggle duration increases with foraging distance in a simple linear fashion (*Schürch et al., 2013*, *2016*, *2019*; *Kohl et al., 2020*). Interestingly, Karl von Frisch, the discoverer of the spatial information content of the waggle dance, and his colleagues had originally proposed a different shape of the function (*von Frisch & Jander, 1957*; *von Frisch & Kratky, 1962*; *von Frisch, 1965*). Based on several early experiments conducted between 1945 and 1960 it was established that the distance communication function is non-linear in that the slope of waggle duration flattens beyond a certain distance (Fig. 1A) (*von Frisch, 1965*). Such a shape was independently corroborated later by a study of *Visscher (1982)*.

To identify possible reasons for this apparent contradiction between the original and the current view, we closely inspected all studies available to us in which Western honey bees (*Apis mellifera*) of European origin were trained to feeders to infer the distance function of the waggle dance (Table 1). This revealed that authors of earlier studies (performed before the 1990s) generally trained the bees to much further distances (usually well beyond 1 km) than authors of more recent studies. In fact, there seems to be an association between the maximum distances the bees were trained, and whether curvature was found in the distance communication function (Fig. 1B). Unfortunately, this relationship is confounded by another variable differing between earlier and more recent studies: authors of earlier studies usually considered a correlate of the distance signal, the duration of whole dance circuits (waggle phase plus return phase), instead of measuring waggle phase duration. This was done mainly because circuit duration can be directly measured with a stopwatch, whereas one needs to record dances on video to precisely measure waggle phase durations, which was expensive before digital video cameras became readily available. Consequently, there are two explanations for the apparent ambiguity concerning the shape of the distance communication function. Perhaps, curvature indeed

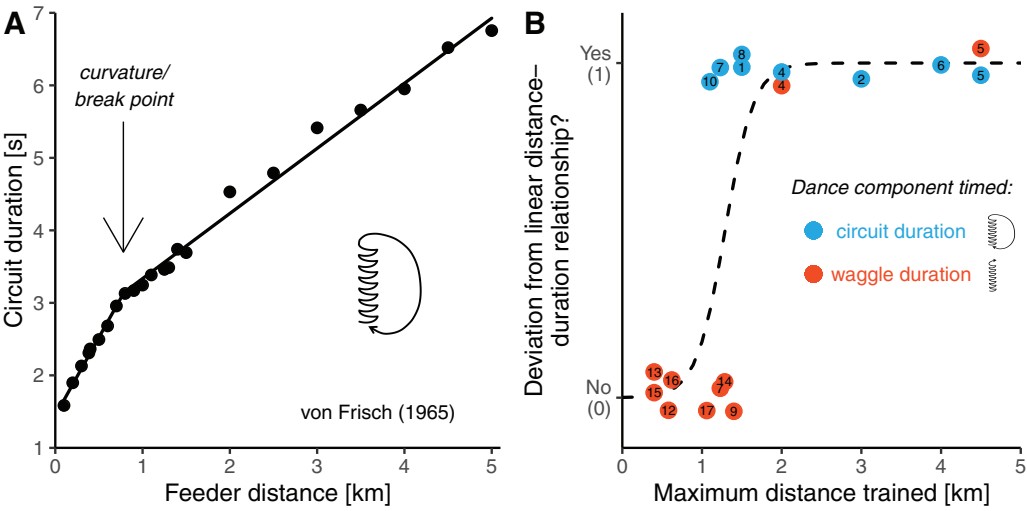

**Figure 1 Potential curvature in the distance communication function.** (A) Dance circuit duration in *A. mellifera carnica* as a function of feeder distance. Data are average values presented by *von Frisch (1965)* in his synopsis on the dance language (original data are up to 9.5 km). The distance-circuit duration relationship deviates from a simple linear shape (the line is a segmented regression with a break-point at 0.7 km distance). (B) Analysis of published feeder training experiments with respect to whether the distance communication function was found to be non-linear (*y*-axis) as a function of how far the bees were trained away from the hive (*x*-axis). Colours indicate whether researchers timed the waggle duration (the actual distance signal, red) or the circuit duration (waggle phase plus return phase, a proxy of the distance signal, lightblue). The dashed line is a logistic regression model. The studies considered are listed in Table 1 (study number eleven is not considered in this figure, and the study by Lindauer (1960) (see *von Frisch, 1965*), who trained bees up to 11 km from the hive is outside of the plotting range of the figure).

exists in the distance-*waggle duration* function, but it can only be clearly detected when the bees are trained significantly further than 1 km from the hive. Alternatively, early studies found curvature in the distance-*circuit duration* relationship only because there is curvature in the relationship between distance and the *return phase duration*, not the waggle phase duration.

There are two historical studies which could have solved the problem since bees were trained to feeders at relatively long distances, and the three durations components of the resulting dances were determined (*von Frisch & Jander, 1957*; *Wenner, 1962*). Unfortunately, these studies seem to have conflicting results. *von Frisch & Jander (1957)* trained bees to feeders at seven distances between 200 and 4,500 m from the hive and made time measurements from film recordings. Their data suggested that there is curvature in the distance-waggle duration function and little or no curvature in the distance-return duration function (Fig. 2). Based on this study, *von Frisch (1965)* concluded that the shape of the circuit duration function well reflects the shape of the waggle duration function. *Wenner (1962)* trained bees to feeders at 16 distances between 31 and 1,230 m from the hive and determined the different duration components from sound recordings. He found that the waggle phase duration followed essentially a straight line, while there was clear curvature in the relationship between distance and return phase duration (Fig. 2). However, it is unclear whether Wenner's results actually contradict the data of von Frisch

**Table 1 Published distance-training experiments with temperately adapted European *A. mellifera*.** The study IDs are the same as used for the numbering of data points in Fig. 1A. Information is provided on the bees used, the location of the experiment, the maximum distance to which the bees were trained, the components of the waggle dance measured, and whether the observed relationship between feeder distance and the duration of the dance component(s) deviated from linearity. The last column contains the references.

| Study ID | Honeybee subspecies | Location | Max. distance trained (km) | Dance component(s) timed | Deviation from linear distance-duration relationship? | References |
|---|---|---|---|---|---|---|
| 1 | *A. m. carnica* | Brunnwinkl, Austria | 1.5 | Dance circuit | Yes | von Frisch 1945 (in *von Frisch, 1965*, p. 66) |
| 2 | *A. m. carnica* | St. Gilgen, Austria | 3 | Dance circuit | Yes | von Frisch 1946 (in *von Frisch, 1965*, p. 66) |
| 3 | *A. m. carnica* | Graz, Austria | 11 | Dance circuit | Yes | Knaffl & Lindauer 1949 (in *von Frisch, 1965*, p. 67) |
| 4 | *A. m. carnica* | Bonn, Germany | 2 | Waggle phase<br>Dance circuit | Yes<br>Yes | *Steche (1957)* |
| 5 | *A. m. carnica* | Munich, Germany | 4.5 | Waggle phase<br>Dance circuit | Yes<br>Yes | *von Frisch & Jander (1957)* |
| 6 | *A. m. carnica* | Munich, Germany | 4 | Dance circuit | Yes | Lindauer 1960 (in *von Frisch, 1965*, p. 66) |
| 7 | *A. m. ligustica* | Pinckney, Michigan, USA | 1.23 | Waggle phase<br>Dance circuit | No<br>Yes | *Wenner (1962)* |
| 8 | *A. m.* sspp.-hybrid | Arnot Forest, New York, USA | 1.5 | Dance circuit | Yes | *Visscher (1982)* |
| 9 | *A. m. ligustica*-hybrid | Japan | 1.4 | waggle phase | No | *Sasaki, Takahashi & Sato (1993)* |
| 10 | *A. m.* sspp.-hybrid | Florida, USA | 1.1 | Dance circuit | Yes | *Waddington et al. (1994)* |
| 11 | *A. m. carnica* | Göttingen, Germany | 1 | Dance circuit | Yes* | *Steffan-Dewenter & Kuhn (2003)* |
| 12 | *A. m.* sspp.-hybrid | Canberra, Australia | 0.58 | Waggle phase | No | *Tautz et al. (2004)* |
| 13 | *A. m. ligustica* | Hangzhou, China | 0.4 | Waggle phase | No | *Su et al. (2008)* |
| 14 | *A. m. mellifera*-hybrid | Brighton, UK | 1.285 | Waggle phase | No | *Schürch et al. (2013)* |
| 15 | *A. m. mellifera*-hybrid | Brighton, UK | 0.4 | Waggle phase | No | *Schürch et al. (2016)* |
| 16 | *A. m. ligustica*-hybrid | Blacksburg, Virginia, USA | 0.622 | Waggle phase | No | *Schürch et al. (2019)* |
| 17 | *A. m. ligustica*-hybrid | Afton, Minnesota, USA | 1.06 | Waggle phase | No | *Carr-Markell et al. (2020)* |

**Notes:**
* The results of *Steffan-Dewenter & Kuhn (2003)* suggest a non-linear relationship of distance and circuit duration, but since bees were tested at a low number of feeder distances (ca. 260, 500 and 1000 m from the hive), the shape of the distance-duration function cannot be unambiguously determined.

and Jander. On the one hand, Wenner's maximum training distance of 1,230 m was perhaps too short to detect the curvature in the waggle duration function. On the other hand, Wenner reports that sugar concentration was gradually rising during his experiment because of hot temperatures. Since honey bees perform quicker return phases when dancing for food sources that are higher in quality (*Seeley, Mikheyev & Pagano, 2000*; *Hrncir et al., 2011*; *Łopuch & Tofilski, 2020*), the gradual increase in sugar concentration could explain why the bees reduced the return phase duration at long feeder distances (that were reached later in the experiment).

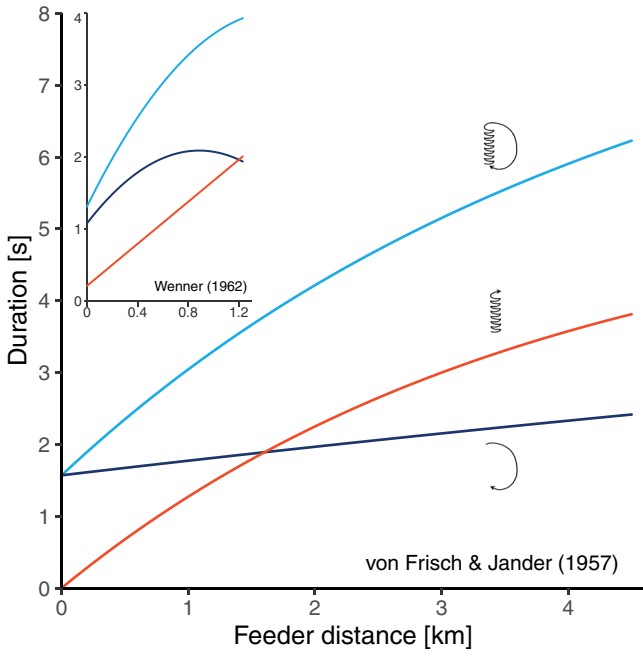

**Figure 2 Results of two historical studies exploring the relationships between feeder distance and the duration of three components of the waggle dance in *A. mellifera*.** Red: waggle phase, dark blue: return phase, light blue: whole dance circuit. The main graph shows the curves obtained by *Von Frisch & Jander (1957)* who trained bees up to 4.5 km from the hive. The small graph embedded in the top left corner shows the curves obtained by *Wenner (1962)* who trained bees up to 1.23 km from the hive. The graphs are scaled equally. In both studies, authors report that the circuit duration function is non-linear, however, while Von Frisch and Jander's data suggest that the curvature stems from the waggle phase, Wenner's data suggest that it stems from the return phase of the dance.

To solve the ambiguity about the distance-duration relationship of the waggle dance, we performed a new experiment. We trained bees to forage at an array of feeder positions up to a relatively long distance of 1.7 km from the hive and determined both the waggle phase duration and the return phase duration of their corresponding dances from video recordings. Then, we fit three types of models to the distance-duration relationships—a linear model, a non-linear model, and a break-point model—and compared their goodness-of-fit to determine which captured the distance-duration relationships best.

## MATERIALS AND METHODS

### Feeder training experiment

We conducted the experiment in the Steigerwald, a forested hill region located between the cities of Würzburg and Nuremberg in Germany. Permits to work in the study area were granted verbally by Ulrich Mergner, head of the Bavarian State forest administration in Ebrach, Germany.

Since honey bees use optic flow to estimate the distance flown to a goal, variation in the visual appearance of the terrain can affect their distance estimations (*Tautz et al., 2004*). Although the visual odometer is thought to be quite robust with respect to the contrast variation the bees typically encounter when foraging (*Si, Srinivasan & Zhang, 2003*; *George*

*et al., 2020*), we wanted to make sure that the effect of foraging distance on dance behavior was not confounded by variation in the terrain. The forest location allowed us to train the bees in a homogenous terrain along a straight gravel road through beech forest. With this setting, the bees encountered the same scenery of mature forest at every section of their flight paths, so that variation in waggle duration between feeder locations was attributable to changes in feeder distance alone.

We worked from 8 August 2019 to 12 August 2019, when natural forage was scarce so that the bees could easily be lured to an artificial food source. Two days before feeder training, we transferred a colony of *A. mellifera carnica* housed in an observation hive to the experimental site (N 49.8618, E 10.5366). The colony was queen right and had brood in all stages. The hive, which we mounted onto a wooden stand, contained two frames stacked over each other (standard Zander frames) and had glass windows on both sides which could be covered by wooden boards. A wedge placed behind the hive entrance forced homecoming bees to enter the colony on one side of the combs. This made it possible to capture most dances while only recording one side of the colony. During video observations, we used a light-proof cloth spread out over the hive to prevent direct sunlight from entering the colony.

To train the bees to feeders at varying distances from the hive we first offered the bees a piece of wax comb filled with concentrated sucrose solution in front of the hive entrance. Once the bees discovered the bait, we transferred the feeder with the bees to a small colored table at knee-height and moved this feeding station step by step until we reached the first test location at 100 m distance (determined with a hand-held GPS device). We let the bees forage for around 30 min to adjust to the location before we started dance observations. During that time, we labelled each individual bee visiting the feeder with a unique combination of shellac color paints on the thorax and/or abdomen. Then we video recorded their dances back at the hive at a frame rate of 50 Hz using a digital camcorder (Panasonic HC-X929). The sequence of moving the feeder, marking bees and video recording was repeated for eight other feeder locations, each 200 m further from the hive up to the final feeder distance of 1.7 km. Throughout the experimental period weather conditions were suitable for honey bee foraging (no rain, temperatures between 18 and 28 °C, mean wind speed of only 2.7 m/s outside the forest; data from the nearest weather station: agrarmetereologie Bayern, weather station Kleingressingen, www.wetter-by.de). At all feeder distances we offered the same sucrose solution (2 M, scented with star anise extract) so that the relative attractiveness of our artificial food source was only affected by the distance to the hive and by potential fluctuations in the colony's parallel intake of other, natural nectar sources. However, we believe that the natural resource environment was largely stable since the bees did not perform many waggle dances for other food sources during the experiment. At the feeder, the bees remained undisturbed by predators or competitors (e.g. wasps or ants) and there was no indication that bees from another colony visited our feeder (i.e. the foragers marked at the feeder showed up at the hive and we did not observe any fights between bees).

## Dance measurements

We first screened the videos and listed all dances of all individually labelled bees that had been recorded. There were 90 dances of a total of 56 identifiable bee individuals captured on the videos. For the analyses, we selected dances in such a way that we represented every bee once while at the same time balancing the number of dances per feeder distance. This resulted in a sample of 56 dances (5–8 dances per distance), each performed by a different bee. Considering that different workers from a single colony vary in their distance communication (*von Frisch, 1965*; *Schürch et al., 2016*), this sampling scheme guaranteed that the distance communication function we inferred was not biased towards any individual bee with extreme behavior.

Following the basic protocol of *Couvillon et al. (2012)* we timed the duration of four consecutive waggle phases per dance, defining waggle duration as the time lapse between the first video frame in which a bee had started shaking its abdomen (identified as the moment when the image of the bee got blurred) until the first frame in which it had stopped waggling again. We also determined the duration of four subsequent return phases (the time lapses between two waggle runs). The measurements were performed manually on a computer using the program utilius fairplay 5 (ccc software, Leipzig, Germany). We averaged to get the mean waggle phase and the mean return phase duration per dance, respectively. By adding mean waggle phase duration and mean return phase duration we obtained the dances' mean circuit duration. These data, one value of waggle phase, return phase and circuit duration per dance, were considered for the analyses.

## Comparisons of distance-duration models

For each of the three time-variables of the dances—waggle phase duration ($t_w$), return phase duration ($t_r$) and their sum, circuit duration ($t_c$)—we tested whether the relationships with feeder distance ($d$) were sufficiently described by simple linear regression models, or whether more complex models (allowing for changes in slope over the range of distances) resulted in better fits. All analyses were performed in R version 3.6.2 (*R Core Team, 2019*). Simple linear regressions, which we refer to as "linear models", were fit using the "lm" function of the stats package. As an alternative model, which allowed for a gradually decreasing slope of duration versus distance and which we refer to as the "non-linear model", we fit a formula proposed by *von Frisch & Kratky (1962)* using the "nls" function from the "stats" package:

$$t = a + \frac{b}{c}\left(1 - exp^{-cd}\right),$$

Frisch and Kratky deduced this formula based on behavioral and physiological considerations, and it well reflected the distance-duration relationships obtained experimentally by *von Frisch & Jander (1957)*. As another alternative to the linear model, we fit a segmented linear regression with two linear equations and a "break-point" using the function "segmented" (from the eponymous R-package; *Vito & Muggeo, 2008*), which we refer to as the "segmented model".
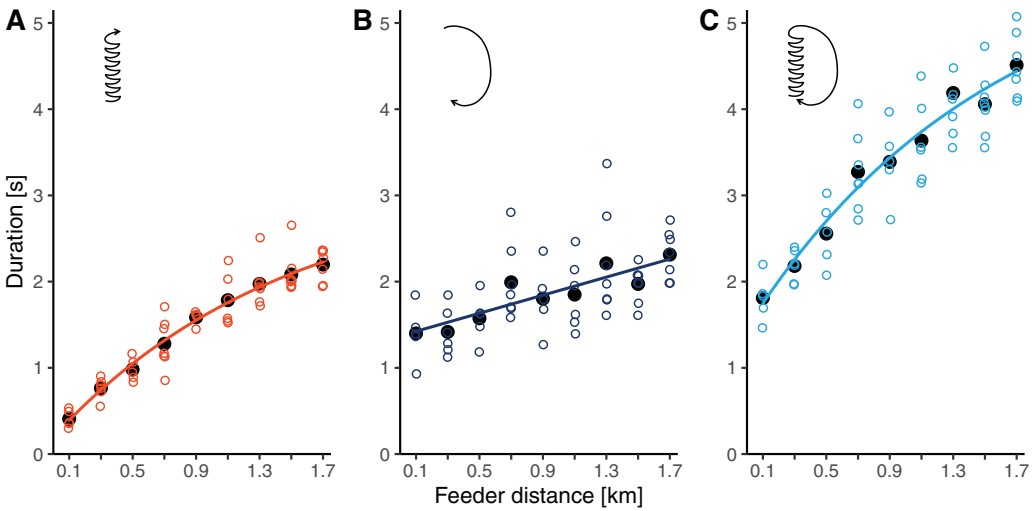

**Figure 3** (A) Waggle phase duration, (B) return phase duration and (C) circuit duration of waggle dances performed by bees trained to feeders at distances between 0.1 and 1.7 km in this study. Open circles are raw data (mean values per dance, each dance by a different bee), solid black dots are mean values per distance, and lines represent the predictions of the best models (based on AICc) for each component (see Table 2).

We evaluated and compared the models based on three criteria. First, we checked whether the predictions showed any biases across the range of feeder distances, i.e. whether the models systematically under- or overestimated the duration components at any point (aided by the inspection of plots of fitted values versus residuals); second, we computed the Pearson's correlation coefficient $r$ for correlations between predicted values and original observations, allowing us to compare how well the models fit the data; and third, we compared the models' Akaike information criterion values (for small sample sizes, AICctab-function of the bbmle package, *Bolker & R Core Team, 2017*), allowing us to check which of the three models would optimize fit versus complexity.

## RESULTS

Waggle phase durations increased by 5.4 times from 0.41 ± 0.10 s at 100 m to 2.20 ± 0.19 s at 1.7 km (throughout this section, results are reported as means ± SDs). The relationship clearly deviated from simple linearity, as the slope flattened with increasing distance from the hive (Fig. 3A). Accordingly, our non-linear model, which allowed for a continuous decrease in slope, had a better fit than the linear model ($r = 0.946$ versus $r = 0.935$, ΔAICc: 7.7). The segmented model, which divided the relationship into two linear regressions separated at a break-point, fit the data similarly well as the non-linear model ($r = 0.947$, ΔAICc: 1.5). The breaking point was estimated to lay at around 1 km distance (Table 2).

Return phase durations also increased with distance but at a slower rate and with more variability than waggle phase durations. At feeder distances between 100 m and 1.7 km, return phase durations rose from 1.40 ± 0.33 s to 2.31 ± 0.29 s (a 1.7-fold increase, Fig. 3B). The relationship was sufficiently described by a simple linear model (Table 2).

**Table 2 Model comparisons for the distance-duration relationships.** Comparison of non-linear, segmented linear and simple linear models of the relationships between feeder distance ($d$, in km) and the duration of the waggle phase ($t_w$), the return phase ($t_r$) and the whole dance circuit ($t_c$) (each in s) based on the data obtained in this study. The models are ordered according to their $\Delta$AICc values (the "best" models are highlighted in bold).

| Dance component | Model | Is is biased?[$] | Formula | $r$[#] | $\Delta$AICc |
|---|---|---|---|---|---|
| Waggle phase | **non-linear** | **no** | $t_w = 0.1993 + \dfrac{2.0018}{0.6717}\left(1 - e^{-0.6717 * d}\right)$ | **0.946** | **0** |
| | segmented | no | $t_w = \begin{cases} 0.2917 + 1.4282 * d, & d \leq 1.0328 \\ 1.0767 + 0.6683 * d, & d > 1.0328 \end{cases}$ | 0.947 | 1.5 |
| | linear | yes | $t_w = 0.4475 + 1.1152 * d$ | 0.935 | 7.7 |
| Return phase | **linear** | **no** | $t_r = 1.3712 + 0.5238 * d$ | **0.582** | **0** |
| | non-linear | no | $t_r = 1.2686 + \dfrac{0.8876}{0.6035}\left(1 - e^{-0.6035 * d}\right)$ | 0.587 | 1.9 |
| | segmented | no | $t_r = \begin{cases} 1.2303 + 0.8613 * d, & d \leq 0.7 \\ 1.5725 + 0.3725 * d, & d > 0.7 \end{cases}$ | 0.596 | 3.4 |
| Dance circuit | **non-linear** | **no** | $t_c = 1.467 + \dfrac{2.893}{0.6519}\left(1 - e^{-0.6519 * d}\right)$ | **0.904** | **0** |
| | segmented | no | $t_c = \begin{cases} 1.4926 + 2.4201 * d, & d \leq 0.7 \\ 2.2844 + 1.2889 * d, & d > 0.7 \end{cases}$ | 0.906 | 1.4 |
| | linear | yes | $t_c = 1.8187 + 1.6390 * d$ | 0.894 | 2.6 |

**Notes:**
[$] Does the model provide biased predictions, i. e. do the predictions systematically deviate from the actual observations at any point over the range of feeder distances? (Decision aided by the inspection of plots of fitted values versus residuals)
[#] Pearson's $r$ of a correlation between fitted and observed values, a measure of how well the models fit the data

Being the combination of the waggle phase and the return phase, circuit duration increased by 2.5 times from $1.81 \pm 0.27$ s to $4.51 \pm 0.37$ s between 100 m and 1.7 km. Given the curvature in the function relating waggle phase duration to distance, the distance related increase in circuit duration was also better described by the non-linear model than by the linear model (Table 2). Again, the segmented model resulted in a fit comparable to that of the non-linear model. For circuit durations, the break-point was estimated to lay at 0.7 km distance.

## DISCUSSION

### Honey bees communicated via a non-linear distance-waggle duration function

By training honey bees to feeders up to a distance of 1.7 km from the hive and by timing their corresponding recruitment dances, we found that their distance function is non-linear in the sense that the increase in waggle duration flattens beyond a certain distance from the hive. While this confirms the original concept of *von Frisch (1965)*, our results are in seeming contrast to more recent studies, who proposed a simple linear relationship (e.g. *Schürch et al., 2013*, *2019*). In the early studies of the waggle dance, researchers trained bees to much further distances (well beyond 1 km) than was done in more recent studies. However, they usually only determined the relationship between distance and the duration of whole dance circuits (waggle phase plus return phase), taking it as a proxy for the bees' distance communication function. This created two problems. On the one hand, it could be that the distance function does not show curvature before a certain threshold distance, so that the non-linearity can simply not be detected when the bees are only trained to feeders at short distances from the hive. On the other

hand it was unclear whether the non-linear shape of the circuit duration function was actually due to non-linearity of the distance function of waggle duration or whether it stemmed from potential curvature in the relationship between distance and the return phase of the dance. Our study solved this ambiguity as we trained bees significantly further than has been done in any of the other recent studies, and as we timed both the waggle phase and the return phase of the dance. This clearly revealed that the distance-waggle duration relationship is non-linear when considered over a large range of distances. However, since the slope of waggle duration only flattens significantly after about 1 km distance from the hive, we also confirmed that the distance-waggle duration relationship is well captured by a simple linear model for a range of closer distances.

Interestingly, the relationship between return phase duration and distance was sufficiently described by a simple linear model. This supports the original assumption of *von Frisch & Jander (1957)* and *von Frisch (1965)* that the curvature detected in the distance-circuit duration curves is attributable to non-linearity in the waggle duration signals. Honey bees tune the duration of the return phase in response to resource quality so that they have quicker return phases when a resource is more profitable (*Seeley, Mikheyev & Pagano, 2000*; *Hrncir et al., 2011*; *Łopuch & Tofilski, 2020*). Therefore, a possible explanation for why mean return phases increased with distance in our experiment is that the bees' ratings of the feeder's profitability lowered while we moved it away from the hive. Since we kept the sugar concentration constant at 2 M, such a reduction in profitability rating would have been due to the longer distances the bees needed to travel to collect the same resource, or due to a gradual saturation of the colony during the experiment. Alternatively, the increasing return phase duration can simply be explained by the distance-related increase in waggle duration. After a waggle run, the bees walk back to their approximate starting point to keep their position on the comb while dancing. Given that the bees walk further stretches on the comb when waggle durations are longer (*Heran, 1956*), they also need to walk back further during the return phase resulting in longer return durations.

Our distance-waggle duration function is similar in shape to a function fit through the historical reference data by *von Frisch & Jander (1957)*, who determined waggle durations at seven feeder distances between 0.2 and 4.5 km (Fig. 4A, the red line and black line). Furthermore, our data on the distance-circuit duration relationship are in good accordance with the data presented by *von Frisch (1965)* (which are averages based on several studies) (Fig. 4B). Especially there is a smooth transition between the values at our furthest training distances and the data presented by von Frisch from 2 km onwards. The general similarity was expected given that the bees used by us and those used by von Frisch and his colleagues belonged to same subspecies of bees.

## The general shape of the distance communication function of honey bees

The insight that the distance-waggle duration function is non-linear and that the form of the *circuit duration* curve represents the form of the *waggle duration* curves allows us to consider the results of both old and new studies to outline the general shape of the distance

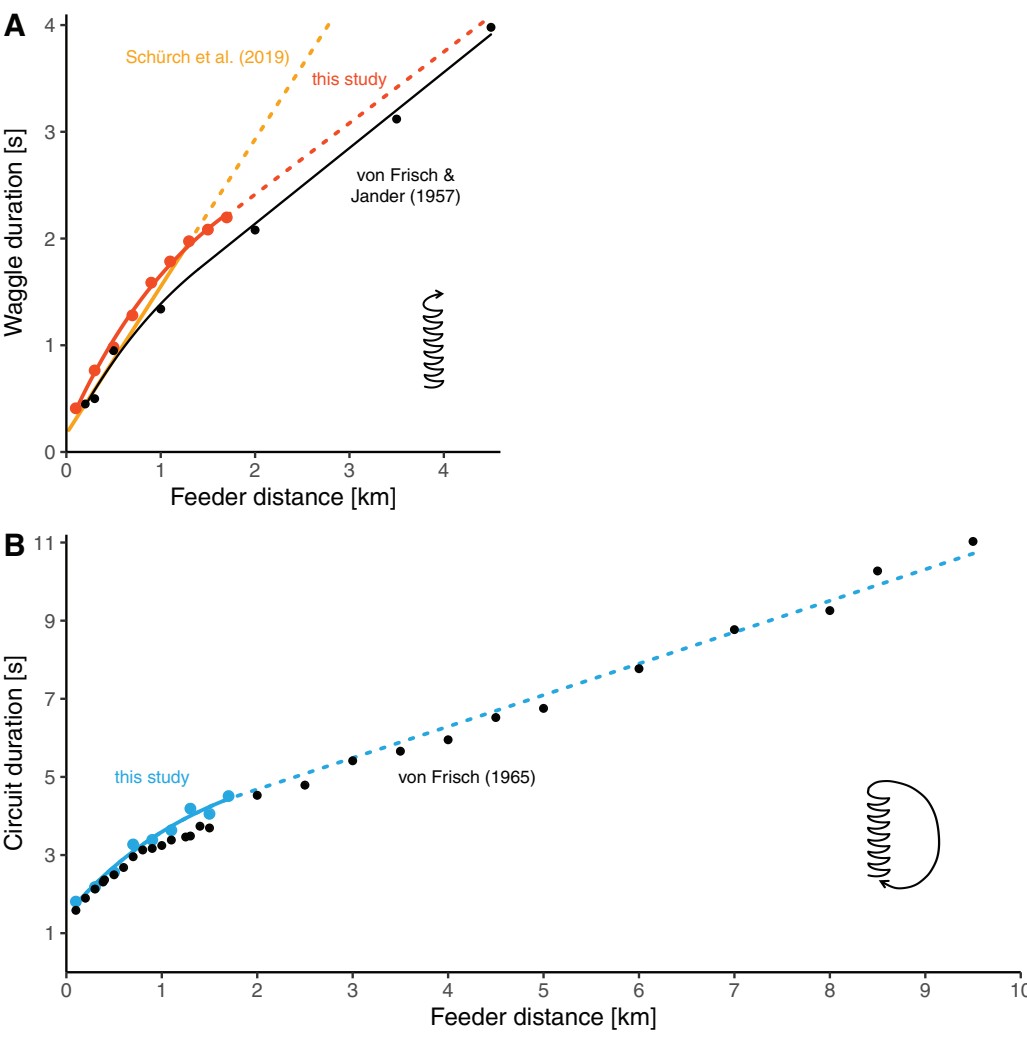

**Figure 4 Comparisons of distance-duration curves.** (A) The relationship between waggle duration and feeder distance obtained in this study (red dots, mean values per distance) compared to the results of *Von Frisch & Jander (1957)* (black dots, mean values per distance). Data from this study are modelled by the non-linear function for the data range up to 1.7 km (solid red line, the same model as presented in Fig. 3A, see Table 2) and extrapolated with the second segment of the segmented model (dashed red line, see Table 2). The data of Frisch and Jander were remodeled by us with a non-linear model up to 1.4 km distance and with a linear model beyond (solid black line; for $d \leq 1.4$ km: $t_w = 0.1096 + 1.7208/0.6272 \times (1-e^{-0.6272 \times d})$, for $d > 1.4$ km: $t_w = 0.7198069 + 0.70947 \times d$). The yellow line represents the linear model presented by *Schürch et al. (2019)* which is based on several feeder training experiments (the solid line segment represent the range of distances up to 1.285 km, for which data were available; the dashed line segment is the extrapolation). (B) The relationship between circuit duration and feeder distance obtained in this study (light blue dots, mean values per distance) compared to data presented by *Von Frisch (1965)* (black dots, average data from several studies per distance). The solid light blue line up to 1.7 km stems from the non-linear model based on our data (the same model as presented in Fig. 3C, see Table 2). The dashed line is a linear extrapolation based on the values presented by Von Frisch (formula: $t_c = 3.072032 + 0.8046 \times d$).

communication function. In referring to it as the "general shape", we mean that it probably applies to the distance functions of all honey bee species, although there are population- and species-specific differences in the slopes of these functions (*Kohl et al., 2020*).

The evidence for this stems from experiments with four non-European honey bee (sub)species: in studying dances of *A. mellifera scutellata* from tropical Africa, *Schneider (1989)* found that in this honey bee subspecies the relationship between distance and waggle duration was better described by a curve with a decreasing slope than by a straight line; and in training foragers of the three Asian honey bee species *A. florea*, *A. cerana* and *A. dorsata* to feeders at relatively long distances in Thailand, *Dyer & Seeley (1991)* found that the distance-circuit duration curves of each of the three species followed a segmented function with the first linear segment being steeper than the second one.

*von Frisch & Kratky (1962)* proposed that the distance-waggle duration function has the shape of a curve with a continuously flattening slope. From a physiological perspective there would be an obvious proximate explanation for such a model. When flying to a foraging site, a honey bee perceives optic flow which she uses to estimate flight distance (*Esch & Burns, 1996*; *Srinivasan, 2000*; *Esch et al., 2001*; *De Marco & Menzel, 2005*). With increasing flight distance, however, the neurons processing the sustained optic flow stimulus might decrease their responsiveness (*Maddess & Laughlin, 1985*; *Clifford & Ibbotson, 2002*; *Taylor & Krapp, 2007*). As a consequence, segments of the flight path that are travelled later during a trip would lead to lower distance estimates than earlier segments. This form of neural fatigue would result into the proposed continuously flattening curve (*von Frisch & Kratky, 1962*). The curve would then be interpretable as the result of a physiological constraint or insufficiency.

However, the available data rather suggest a different scenario, namely that the distance-waggle duration function has two linear phases—a steeper phase for closer distances and a lower phase for longer distances—mediated by a rather sudden, curved transition around a threshold distance. There are several indications that such a segmented relationship captures the process better than a continuously flattening slope. First, all studies in which bees were trained to relatively short distances (up to around 1 km distance) found that the relationship was well described by a simple linear model (*Sasaki, Takahashi & Sato, 1993*; *Tautz et al., 2004*; *Su et al., 2008*; *Schürch et al., 2013*, *2016*, *2019*; *Kohl et al., 2020*). Second, the data from studies in which bees were trained well below 1 km suggest that waggle duration (or circuit duration) increases linearly over the range of far distances from about 1 km onwards (*von Frisch & Jander, 1957*; *von Frisch, 1965*) (Figs. 4A and 4B). Third, the average data on the distance-circuit duration relationship which are based on several studies compiled by *von Frisch (1965)* suggest a sudden break-point rather than a continuously flattening slope (Figs. 1A and 4B). And fourth, our data were fitted equally well by a segmented model as with a non-linear continuous curve (Table 2). Of course, it is hard to believe that a behavioral process shows a clear-cut break-point. Even if individual bees possessed distance-curves with such a sudden break-point, variation between individuals of a colony and variation between colonies of a populations would lead to a function with a "smooth" turning point on the colony or population level, respectively. The segmented function seems to be a good *model*, though. It suggests that the bees process distance differently before and beyond a certain threshold distance. While the physiological basis for this might well involve a sensory adaptation

process, a segmented relationship could not simply be explained by a mere physiological constraint. Instead, this shape suggests that it has some adaptive value.

Dancing honey bees have the problem that they are limited in how long waggle phases can potentially be since very long waggle runs become difficult to follow by potential recruits and take a lot of time during which the dancers cannot collect food themselves. Consequently, they face a trade-off between the steepness of their distance-waggle duration functions (which affects the communication precision) and the maximum foraging distances which they can indicate in their dances (*Dyer, 2002*; *Kohl et al., 2020*). In line with this, honey bee (sub)species with smaller foraging ranges evolved steeper distance-duration functions (*Kohl et al., 2020*). However, the segmented distance function could reduce some pressure from this trade-off between communication precision and communication range: with a segmented distance function, the bees would have a first steeper segment to communicate more precisely over a range of closer distances from their hive, while the second lower segment would still allow them to communicate foraging sites at far distances, if necessary. As an extension of the adaptive-tuning hypothesis for the evolution of distance dialects (*Dyer & Seeley, 1991*; *Kohl et al., 2020*), the prediction is that honey bee populations with longer foraging ranges have distance functions with break-points at further distances. The available data are in line with this prediction: temperate honey bees, which have larger foraging ranges than tropical honey bees (*Kohl et al., 2020*), also have break-points at further distances (~1 km) than tropically adapted honey bees (break-points between 200 and 560 m, *Schneider, 1989*; *Dyer & Seeley, 1991*).

## Caveats

Before reasonable analyses of the proximate and ultimate causes can be performed, the apparently non-linear distance communication functions need to be explored and described in more detail. For example, even though we trained bees further away from the hive than it was done in any other study since the experiment of Lindauer in 1960 (see Table 1), our maximum training distance of 1.7 km was clearly not long enough to unambiguously determine the shape of the distance function at long distances because temperate honey bees are known to forage at distances more than 10 km away from the hive (*Beekman & Ratnieks, 2000*). The question whether the distance function continues linearly or progressively flattens over the range of long foraging distances beyond around 1 km, however, is crucial for our biological interpretation of the function (see above). Furthermore, the distance-curve we inferred from our experiment is a representation of how distance is communicated on the level of the *colony* because we analyzed dances that were each performed by a different bee. We do not know whether the distance-curves of *individual bees* look the same. An alternative explanation, namely that the non-linear shape of the function on the colony level arose because bees that preferred to forage at closer distances had steeper distance-curves than bees that preferred to forage at longer distances, needs to be excluded. This is crucial given that any inferences about how the bees process distance can only be made based on the distance communication of individual bees. While it was demonstrated in several early studies that the bees can indeed be trained to much further distances (Table 1), having *the same bees* dancing for an array of feeders over a

longer range of distances is challenging. In fact, moving a sugar feeder further usually results in a high turnover of foragers. For example, of the 56 bees which we recorded dancing in our experiment, 42 only danced for one feeder (75%), 10 danced for two feeders (18%) and only four danced for three feeders (7%). To ensure that the same group of bees dances for feeders at several distances, the feeder needs to be moved slowly, ensuring that all bees can keep up. On the other hand, training a colony of honey bees to very far distances, requires quickly moving the feeder forward (*von Frisch, 1965*). Some experimental tuning needs to be done in future studies to overcome this trade-off.

## Implications for waggle dance decoding

Irrespective of the contribution to basic research, our results have direct implications for an ecological application of the waggle dance. Eavesdropping on waggle dances and decoding their spatial information of is a unique tool to study where honey bee colonies collect forage, and thus to explore how key generalist flower visitors evaluate the landscape (*Visscher & Seeley, 1982*; *Couvillon, Schürch & Ratnieks, 2014*; *Wario et al., 2017*; *Young et al., 2021*). To infer the distances of unknown food sites, researchers have measured either circuit duration (e.g. *Visscher & Seeley, 1982*; *Beekman & Ratnieks, 2000*; *Steffan-Dewenter & Kuhn, 2003*; *Danner et al., 2016*; *Park & Nieh, 2017*) or waggle duration (e.g. *Couvillon et al., 2015*; *Bänsch et al., 2020*; *Carr-Markell et al., 2020*) from dances. The use of circuit duration has the advantage that it is faster since it requires only a single time measurement of a sequence of consecutive circuits per dance to obtain a dance's mean circuit duration, and that it can be determined by direct observation and timing in the field. Furthermore, since Karl von Frisch and his colleagues established an extensive data basis of how circuit duration increases with foraging distance (at least for temperately adapted Carnolian honey bees, *A. m. carnica*, Fig. 4B, *von Frisch, 1965*), one can accurately infer the foraging distances of these bees over a long range of potential distances. However, the duration of whole dance circuits includes the duration of the return phase of the dance which is quite variable and conveys little information with respect to foraging distance (Fig. 3B). Furthermore, the return phase duration introduces additional variation since it is adjusted by the bees according to resource quality (*Seeley, Mikheyev & Pagano, 2000*; *Hrncir et al., 2011*; *Łopuch & Tofilski, 2020*). Hence, to enhance the precision of foraging distance estimation, it became standard procedure to measure only the actual distance signal of the bees, waggle duration, from dances (*Couvillon et al., 2012*). Awkwardly, studies that used waggle duration to infer foraging distance relied on linear calibration functions that were experimentally established only over a relatively short range of feeder distances and which did not capture the curvature of the distance-duration relationship (as outlined above). The consequence is that the foraging distances indicated by dances with waggle durations longer than about 1.5–2 s (indicating sites further than about 1–1.5 km distance from the hive) were probably systematically underestimated. For example, *von Frisch & Jander (1957)* determined that the mean waggle duration of dances of *A. mellifera carnica* was around 4 s when foraging at a feeder in 4.5 km distance, but a waggle duration of 4 s translates into a foraging distance of only 2.8 km when

using a recently published distance-waggle duration calibration (*Schürch et al., 2019*). This inaccuracy is crucial given that typically more than 50% of foraging in temperate *A. mellifera* takes place beyond 1 km distance from the hive (*Visscher & Seeley, 1982*; *Beekman & Ratnieks, 2000*; *Steffan-Dewenter & Kuhn, 2003*; *Couvillon et al., 2015*). We recommend that researchers who plan to do forage mapping should calibrate the distance-waggle duration relationship of their bees by training them to an array of feeder up to distances of at least 1.5 or 2 km, so that the relationship beyond the break-point at around 1 km can be unambiguously established. We also recommend generating a calibration specific to the bees of interest because different populations and species of honey bees can differ in their distance communication functions (*Bosch, 1957*; *Kohl et al., 2020*). Such a training experiment (performed during times of little natural nectar flow) will take only about a week time, which is a small fraction of the whole duration needed to complete a forage mapping study. Further, more long-range feeder training studies will lead to a better general understanding of the bees' distance communication function.

## CONCLUSIONS

Based on a number of early distance-training experiments, Karl von Frisch and his colleagues concluded that the relationship between foraging distance and waggle duration in the waggle dance of the Western honey bee is well described by a curve with a decreasing positive slope. However, later studies repeatedly found that waggle duration increases linearly with distance, i.e. as a straight line, not as a curve. We suspect the results of these later studies are only applicable to distances less than about 1 km, since they are based on experiments in which bees were trained to feeders at relatively short distances. Training bees to longer distances revealed that the distance-related increase in waggle duration is better described by a curve or by a segmented model with two linear segments than by a simple linear regression. Our results can directly be applied to improve the inference of foraging distances based on waggle dance observations. *Why* the bees' distance communication function has a non-linear shape is a remaining question.

## ACKNOWLEDGEMENTS

We thank I. Steffan-Dewenter and S. Schiele for their support and for providing the honey bee colony and the equipment used for the experiment. We are grateful to two anonymous reviewers whose valuable comments helped to improve this paper.

### Funding
The authors received no funding for this work.

### Competing Interests
The authors declare that they have no competing interests.

## Author Contributions

- Patrick L. Kohl conceived and designed the experiments, performed the experiments, analyzed the data, prepared figures and/or tables, authored or reviewed drafts of the paper, and approved the final draft.
- Benjamin Rutschmann conceived and designed the experiments, performed the experiments, authored or reviewed drafts of the paper, and approved the final draft.

## Field Study Permissions

The following information was supplied relating to field study approvals (i.e., approving body and any reference numbers):

Permits to work in the study area were granted verbally by Ulrich Mergner, head of the Bavarian State forest administration in Ebrach, Germany.

## Data Availability

The raw data and script are available in the Supplemental Files.

## Supplemental Information

Supplemental information for this article can be found online at http://dx.doi.org/10.7717/peerj.11187#supplemental-information.

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
