# Peer review of "Honey bees communicate distance via non-linear waggle duration functions"

_PeerJ, doi:10.7717/peerj.11187_

## Round 0.1 · original submission · Major Revisions

Dear Drs. Kohl and Rutschmann:

Thanks for submitting your manuscript to PeerJ. I have now received two independent reviews of your work, and as you will see, the reviewers raised some concerns about the research (mostly the manuscript format and content). Despite this, these reviewers are optimistic about your work and the potential impact it will have on research studying honey bee communication. Thus, I encourage you to revise your manuscript, accordingly, taking into account all of the concerns raised by both reviewers.

Please revise your manuscript for clarity and limit jargon/reduce verbiage. Focus on clarity, and address the sections considered incomplete and/or unclear by the reviewers. It appears that certain key references are missing. The Methods should be clear, concise and repeatable. Please ensure this. Also, elaborate on the discussion of your findings, placing them within a broad and inclusive body of work by the field. Please supply your code in the supplemental material.

While the concerns of the reviewers are relatively minor, this is a major revision to ensure that the original reviewers have a chance to evaluate your responses to their concerns. There are not too many suggestions; thus, it should not take much effort to address these concerns to greatly improve your manuscript.

I look forward to seeing your revision, and thanks again for submitting your work to PeerJ.

Good luck with your revision,

-joe

Reviewer 1 ·

Basic reporting

-Is line 41 missing the word “bee”?
-Line 40 and 41. There should be a full description of the dance itself and how is interpreted to convey resource coordinates. For example, how the orientation plays a role in location information. I wonder if this can be added as a graphical abstract? (See [1] for an example).
-Line 67-69. This sentence does not make sense.
-Paragraph beginning at 99. This paragraph needs to be improved in clarity. Example. “To settle this ambiguity in the dance-distance relationship, we trained bees to visit feeders located at varying distances greater than 1km. We then determined the duration of the waggle dance components: waggle phase and return phase.” Then add how this data was analyzed to show the non-linear relationship.
-Line 126. Spelled shellac.
-Results Section. It will improve readability if the results are stated to be reported as mean ± std, then simply report the numbers without restating mean±std every time the numbers are given. “We report results as mean±std.” Subsequent results are just “2.20±0.19”.
-Line 401. Please consider the verbiage. “We suspect these later studies to be limited due to the shorter distance of the resource and thus their results are applicable to distances less than 1km.”

[1] Automatic detection and decoding of honey bee waggle dances
Wario F, Wild B, Rojas R, Landgraf T (2017) Automatic detection and decoding of honey bee waggle dances. PLOS ONE 12(12): e0188626. https://doi.org/10.1371/journal.pone.0188626

Experimental design

-Can the authors add a sentence or two describing how the quality of the resource (feeder) was maintained for the experiment, given that this affects dance performance? This is indicated in the discussion but having it here will quash doubts early on in a reader.
-Could the authors also clarify local feeder conditions (i.e. heavy wind, dim sunlight, competing foragers from native hives, hornets)? Adverse feeder conditions will decrease the bee’s perception of the food source and can affect the return duration.
-I am also curious to know if there were tremble dances from the painted bees as the distance increased. Tremble dances are known to occur in order to inhibit recruitment to a resource.

Validity of the findings

The data analysis appears straightforward and the csv file is readable.

Additional comments

-Citation format seems to randomly change (i.e. line 207).
-Might be a good idea to join the Figure 2 plots. Since the distance is not scaled the same per plots, they look comparable on first sight when they are really not. Perhaps try dashing either von Frisch’s data or Wenner’s data so they can be easily distinguished if joined?


This is a simple yet informative study that clarifies the relationship between waggle dancing and the distance information it conveys. I welcome this study as it clarifies suspicions myself and colleagues had at the time. We suspected the location information to be conveyed in a logarithmic fashion (in line with the flattening theory) but this study suggests a sharper break in distance information. This is very useful information for decoding foraging sites and when interpreting dance information from training experiments.

I would like the authors to seriously reread and revise the verbiage in this paper. For instance, line 90 could be simply written as “It is unclear whether Wenner’s results contradicts the data of von Frisch and Jander, as Wenner’s distance of 1230 m could be too short to detect the waggle duration-distance curvature.”

Reviewer 2 ·

Basic reporting

The article is well-written, well-structured, and flows smoothly from introduction to conclusion.

The only very minor issues I noticed were two sentences that were a little unclear:
Line 344: "When the same group of bees shall be observed dancing for feeders at several distances"- was the meaning that "To ensure that the same group of bees dance for feeders at several distances"?
Lines 365-366: " and little informative with respect to distance"- was the meaning here to say that the return phase duration doesn't convey much information about the distance advertised?

And a few typos:
Line 209: "it was done" -> "was done"
Line 218: "it has been done" -> "has been done"
Line 340: "feeder" -> "feeders"
Line 342: "dances" -> "danced"
Line 352: "Eavesdropping waggle dances" -> "Eavesdropping on waggle dances"
Lines 387-388: "Making an own calibration for the local bees is anyway recommended" -> I would suggest: "We also recommend generating a calibration specific to the bees of interest"

Experimental design

You clearly illustrated why it is important to know whether the waggle run duration-distance function is linear. You presented a thorough review of the literature on the subject, including helpful figures and tables to show comparisons between different studies. Figures 1, 2, and 4 were quite interesting, and figure 3 was nicely-formatted to highlight the different functions for waggle phase, return phase, and circuit durations. The methods section is clear and the methods seem appropriate.

I have two minor suggestions, although these changes are not necessary for publication. First, I would suggest saying something more explicit about optic flow likely experienced along the route to the feeders in this study. Of course, there is still uncertainty about the degree to which different landscapes affect the optic flow experienced by bees during their flights. Also, you did say that all feeders were along homogeneous terrain (a straight gravel road through beech forest). However, you might add a sentence to more specifically address the potential consequences for optic flow as some readers will wonder about it. Second, it would be helpful to include the R code that you used in the supplementary materials. It was very helpful that you included a csv of your data.

Validity of the findings

Your findings are interesting and potentially have significant consequences for basic science questions about the evolution of the waggle dance and practical questions about how to accurately map waggle dances. I think you did a good job of acknowledging the limitations of your study, considering the relatively small sample size, the fact that you could not examine differences in the duration-distance calibration among bees, the range of distances that you examined, and the possibility that these findings may not be true for all subspecies of Apis mellifera or species of Apis. However, as you illustrated nicely in your figures using the data from von Frisch and Jander (1957), there are multiple studies pointing to a similar conclusion. Your findings clearly indicate the importance of having more site- and population-specific calibration datasets that include greater distances.

Additional comments

Thank you for revisiting this question. It was clearly needed. I enjoyed reading the paper. You wrote a very thoughtful discussion of the implications of your results, and your figures are both helpful and elegant. I look forward to seeing further work on this question with other populations of bees and with an even greater range of distances.

---

## Round 0.2 · accepted · Accept

Dear Drs. Kohl and Rutschmann:

Thanks for revising your manuscript based on the concerns raised by the reviewers. I now believe that your manuscript is suitable for publication. Congratulations! I look forward to seeing this work in print, and I anticipate it being an important resource for groups studying honey bee communication. Thanks again for choosing PeerJ to publish such important work.

Best,

-joe

Reviewer 1 ·

Basic reporting

The authors have went above and beyond in addressing my concerns in this section. The writing has improved and is far more clear.

Experimental design

They have addressed my primary concern regarding feeder conditions.

Validity of the findings

No comment.

Additional comments

I think the authors have done a great job in improving an already good paper. They met my concerns and went above my expectations in addressing them. Well done!

Reviewer 2 ·

Basic reporting

All comments were addressed by authors

Experimental design

All comments were addressed by authors

Validity of the findings

No comment

Additional comments

You did an excellent job of clarifying the text and methods, including adding references and sharing your R code, which is very helpful. It is an interesting study, and I look forward to seeing it published.